# Regenerative Endodontics as the Future Treatment of Immature Permanent Teeth

Justyna Zbańska [1,*], Katarzyna Herman [1], Piotr Kuropka [2] and Maciej Dobrzyński [1,*]

[1]  Department of Pediatric Dentistry and Preclinical Dentistry, Wroclaw Medical University, Krakowska 26, 50-425 Wroclaw, Poland; katarzyna.herman@umed.wroc.pl
[2]  Division of Histology and Embryology, Wroclaw University of Environmental and Life Sciences, Norwida 25, 50-375 Wroclaw, Poland; piotr.kuropka@upwr.edu.pl
*  Correspondence: justyna.zbanska@umed.wroc.pl (J.Z.); maciej.dobrzynski@umed.wroc.pl (M.D.); Tel.: +48-50-8293-345 (J.Z.); +48-71-7840-361 (M.D.)

**Abstract:** The regenerative endodontic procedure (REP) is an alternative solution for endodontic treatment of permanent teeth with incomplete root apex development. It results in angiogenesis, reinnervation, and further root formation. Indications for REP include immature permanent teeth with necrotic pulp and inflammatory lesions of the periapical tissues. The main contraindications comprise significant destruction of the tooth tissues and a lack of patient cooperation. We distinguish the following stages of this procedure: disinfection of the canal, delivery of the REP components, closure of the cavity, and follow-up appointments. For effective canal disinfection, the use of both rinsing agents and intracanal medicaments is suggested. Sodium hypochlorite and triple antibiotic paste are used most commonly. Light-activated disinfection is proposed as an alternative method. The prerequisite for the regeneration process of the pulp is the supply of its essential components: stem cells, growth factors, and scaffolds to the canal lumen. Blood clotting, platelet-rich plasma, and platelet-rich fibrin are used for this purpose. For a proper course of REP, it is also necessary to close the tooth canal tightly. For this purpose, mineral trioxide aggregate (MTA), tricalcium silicate (Biodentine), or types of glass ionomer cement are employed. The patient should attend regularly scheduled follow-up appointments and each time undergo a thorough interview, physical and radiological examination. The most important indicator of a successful REP is the continued growth of the root in length and thickness and the closure of the root apex visible on X-rays. Many different proposals for a management protocol have been published; the following paper proposes the authors' original scheme. Regenerative endodontics is the future of the endodontic treatment of immature permanent teeth; however, it still requires a lot of research to refine and standardize the treatment protocol. The application of tissue engineering methods seems to be promising, also for mature teeth treatment.

**Keywords:** regenerative endodontic procedure; revascularization; apexogenesis; apexification; immature permanent teeth; regenerative endodontics

## 1. Introduction

Conventional endodontic treatment of permanent teeth with incomplete root development is impossible, due to the significant risk of complications, including root fracture (the walls are thin and the roots are short) or the accidental injection of fluids or filling materials beyond the wide root apex. In the endodontic management of immature teeth, two methods are generally accepted: keeping all or part of the pulp alive, allowing the root to develop naturally (apexogenesis), or—if the pulp is non-vital—stimulating the formation of a hard barrier in its apical part, using appropriate substances inserted into the canal (apexification) [1,2]. The apexification method has many disadvantages. When calcium hydroxide is applied, repeated placement of intracanal medicaments is necessary, which carries the risk of reinfection, and thorough instrumentation inside the root canal

may cause weakening of the canal walls [3,4]. The application of new materials that are an alternative to calcium hydroxide, e.g., MTA or Biodentine, eliminates the problem of intracanal medicament replacement. However, none of these substances stimulates further root development. They only constitute a mechanical barrier, closing the root canal lumen. Thus, treatment is a filled root canal with wall thickness and length developed naturally before pulp necrosis. However, some reports of apical papilla survival after REP in humans and dogs may indicate that root regenerative processes are possible [5,6]. However, in most cases, the tooth does not show adaptation for long-term functioning, as there is a risk of root wall fracture; the greater the risk, the earlier the stage of its development at the time of pulp damage. According to Danwittayakorn et al. [7], this complication occurs with equal frequency in restorations reinforced with a root-canal post as well as with composite material alone.

The regenerative endodontic procedure, proposed in 2004 by Banchs and Trope [8], currently attracts great interest. In the literature, one can find interchangeably used terms, such as revascularization or revitalization. These names stand for biological procedures to stimulate the restoration of tooth tissues, including the pulp–dentine complex and root formation [9]. For this process to occur, three essential components must be present in the root canal: a scaffold for newly forming tissue, stem cells, and growth factors [10]. The key processes that occur in the root canal after regenerative procedures are angiogenesis, reinnervation, and the differentiation of cells responsible for root development [11]. This leads to the formation of vital pulp-, bone-, cementum- or periodontal-like tissue in the canal and thus, root formation [12–15]. The expected therapy outcome is a tooth whose anatomical structure and functional capacity are close to physiological ones [16,17]. In addition, there are numerous reports in the literature about the effectiveness of REP for immature teeth with necrotic pulp [17–19].

This study aims to review the literature and present the latest methods of regenerative endodontics, including the results of research, and propose a possible treatment protocol.

## 2. Materials and Methods

Permanent teeth with uncompleted root development and non-vital pulp are the main indications for REP. Teeth with inflammatory lesions of the periapical tissues may also qualify for it. It was demonstrated that the stage of root development and the width of the apical opening, and the age and general health status of the patient influence therapy outcome [1]. Due to the wide, apical opening, the thin root walls, and the high risk of failure of the apexification procedure, REP treatment is indicated primarily for teeth in stages 1 (<1/2 developed root), 2 (1/2 developed root), or 3 (>1/2 developed root) of root development, according to Cvek [20]. In an almost-developed root with an open apex (stage 4), apexification treatment or REP should be considered. Studies show that REP is possible if the diameter of the apical foramen is <1 mm [21,22]. The prevailing view in the literature is that the minimum apical width at which REP can be performed is 0.5–1 mm [23].

Since the expected treatment outcome is living tissue in the canal, restoration with root–crown posts is excluded. Therefore, REP should not be performed in cases of significant destruction of the hard tissues of the tooth [23].

The highest probability of therapeutic success occurs in young patients between nine and 18 years of age; the younger the patient, the higher the chances of successful therapy [22,23]. It is also vital to establish good cooperation with the patient and the caregivers so that they are adequately motivated to follow hygiene recommendations and attend the follow-up appointments [1].

The relationship between the etiology of pulp necrosis (caries, trauma, malformation) and treatment outcome is yet to be conclusively proven [24].

Many protocols for regenerative treatment can be found in the literature. In all of them, the following steps can be distinguished: disinfection of the canal, delivery of REP components, closure of the canal and cavity, and follow-up visits.

This paper focuses on comparing the REP protocols available in the literature. The selected publications contain a thorough description of different approaches to the root canal disinfection procedure, scaffold type, final tooth restoration material, follow-up appointment frequency, and treatment results. Figure 1 presents a flowchart, which demonstrates the database search strategy used in this paper. We started scoping the available literature in PubMed.gov, using the keywords 'pulp revascularization' or 'regenerative endodontics'. Another filter used was the year of publication—we selected papers written after 2007; then, the search was narrowed to the literature published in English, describing human species and with text availability: free full text or full text. Subsequently, we selected the papers in which the authors accurately describe the disinfection protocol, scaffold and restorative material as well as follow-up visit schemes. However, all of the selected papers describe successful cases. Unfortunately, very few reports on REP failures are available in the literature; thus, the correct choice of the REP protocol and the profound interpretation of its results are significantly hindered. Therefore, the authors of this paper also propose an original REP protocol based on the available literature.

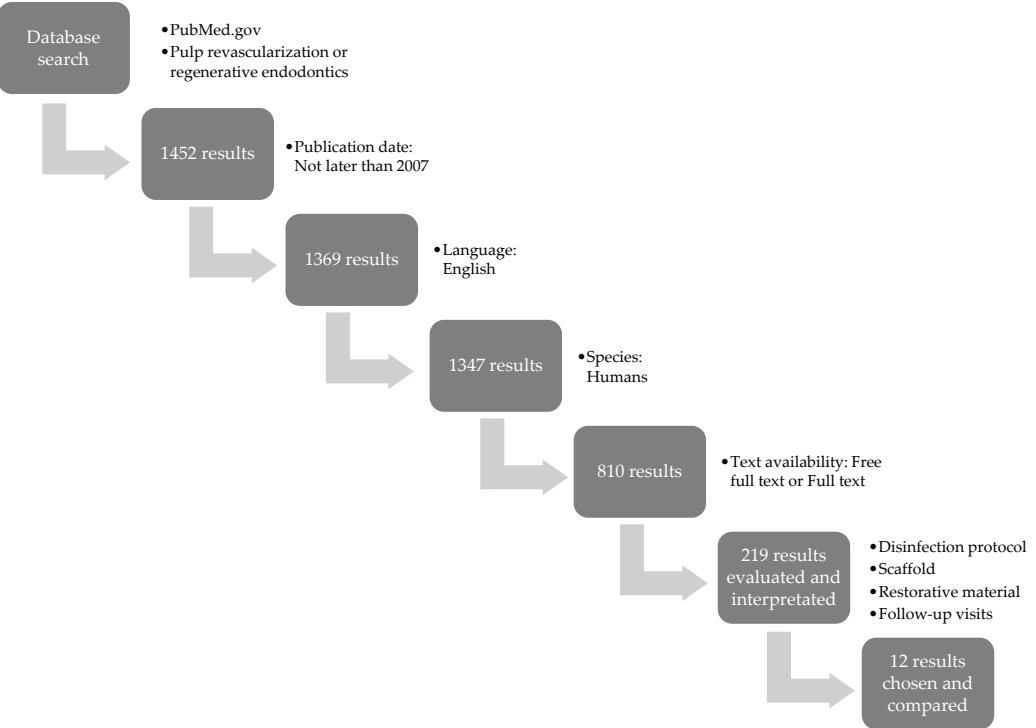

**Figure 1.** Flowchart presenting database search strategy.

## 3. Results

### 3.1. Root Canal Disinfection

One of the main conditions for successful REP is disinfection of the root canal [25]. Leaving microorganisms on the canal walls or in the dentinal tubules prevents the development of new tissue and creates a risk of periapical tissue inflammation [26].

The fundamental way to remove necrotic mass and microorganisms from the canal is to irrigate it with disinfectants [27]. In addition, other methods of disinfection are proposed. Mechanical instrumentation is generally contraindicated to prevent the further weakening of thin canal walls [28]; however, according to some authors, careful preparation of the canal with hand instruments may be necessary to remove bacterial biofilm altogether [29].

During rinsing, gentle manipulation with an endodontic needle (with side holes and a blunt tip) is recommended; it should be loosely placed in the canal lumen, approximately 1–2 mm from the root apex. Careful irrigation with plenty of fluids is required. The primary agent is sodium hypochlorite (NaOCl) of which the concentration used by various authors

varies between 1% and 6% [30]. According to some researchers, high concentrations are not advisable, due to the potential risk of damaging stem cells in the periapical region [11,31]. It was shown that an increase in the concentration of sodium hypochlorite improves the bactericidal effect but at the same time hinders cell proliferation [32–34]. According to the American Association of Endodontists, NaOCl at a 1.5–3% concentration is a reasonable compromise between the antibacterial efficacy of the preparation and stem-cell protection [35]. Heating 2.5% NaOCl to 37 °C results in its disinfecting potential reaching the same level as the 5.25% solution [36]. Alternatively, 0.12% or 2% chlorhexidine (CHX) is used [9], but there is a risk of stem-cell damage, even in this case. Therefore, Widbiller et al. [37] suggested shortening the irrigation time.

After the application of disinfectants, it is suggested to use a 0.9% saline solution (NaCl) to remove the remains of previously used agents, or 17% ethylenediaminetetraacetic acid (EDTA) [38]. The use of EDTA is justified because of its ability to remove the smear layer, resulting in better penetration of the topically applied therapeutic agents. In addition, the chelating effect after EDTA application promotes the release of growth factors from dentinal tubules, migration and differentiation of stem cells, and adhesion of newly forming tissue to the root canal walls [11,29,39].

In addition to rinsing fluids for disinfection, the use of intracanal medications is recommended between visits, for instance, standard triple antibiotic paste (TAP) or calcium hydroxide. TAP consists of minocycline, ciprofloxacin, and metronidazole in a 1:1:1 ratio and an appropriate carrier (propylene glycol, macrogol, saline solution). The choice of antibiotics is dictated by the spectrum and mechanism of action. It was proven that such a combination of drugs guarantees effective antibacterial activity against microorganisms present in the root canal [40,41]. However, several disadvantages were also reported when TAP was used, such as cytotoxicity against stem cells, risk of tooth discoloration, development of drug resistance, and difficult removal from the canal [42,43].

The American Association of Endodontists (AAE) recommends a paste concentration of no more than 1–5 mg/mL, arguing that this would minimize any possible toxic effect on stem cells [35].

A different view is presented by the European Society of Endodontology (ESE) [44,45]. Considering the reported disadvantages of TAP, ESE suggests the use of calcium hydroxide, whose antibacterial efficacy is comparable to TAP, while its toxicity to stem cells is significantly lower [46].

Consensus is yet to be reached among researchers on disinfectants used in REP, and conflicting views can be found in the literature. Brogni et al. [46] described a case of REP failure after TAP application. Repeat treatment with calcium hydroxide (Ca(OH)$_2$) and CHX as irrigating agents was successful. However, some researchers pointed out the risk of weakening the canal walls [44], a possible decrease in the antibacterial properties of calcium hydroxide in contact with inflammatory exudation in the root canal [47], risk of periapical tissue and stem cell damage due to high pH, and the possibility to induce mineralization in the root canal lumen [1]. Bose, comparing the effects of TAP, Ca(OH)$_2$, and formocresol as medicaments used in the endodontic treatment of immature permanent teeth, demonstrated the superior efficacy of TAP in promoting further root development [48]. According to some authors, TAP is biocompatible [49]. Furthermore, minocycline is proven to be non-cytotoxic, inhibit collagenases and metalloproteinases, and increase the anti-inflammatory cytokine Il-10, whereas metronidazole and ciprofloxacin can stimulate fibroblast formation [50]. To reduce the risk of minocycline-induced crown discoloration, the paste should be applied very carefully into the canal so that it does not reach above the cementoenamel junction (CEJ) [44]. Some authors suggest the use of a self-etching bonding system to seal the dentinal tubules in the pulp chamber prior for paste application [51], the alternate use of amoxicillin [52] or cefaclor [53], or a dual-antibiotic paste (DAP) [54].

However, calcium hydroxide, used as an intracanal disinfecting dressing, also has many well-studied advantages. It was shown that the use of calcium hydroxide in combination with EDTA induces the release of TGF-1 and does not adversely affect the survival of

stem cells from the apical papilla (SCAP), as is the case with TAP and high concentrations of NaOCl. On the contrary, it was demonstrated that by releasing growth factors, calcium hydroxide, embedded into dentine walls after irrigation, indirectly affects SCAP survival and proliferation [42,55,56]. It is also much easier to rinse it out completely from the root canal compared to TAP—it is speculated that, despite thorough irrigation, up to 80% of the triple antibiotic paste may remain in the root canal and still negatively affect stem cell survival [42,57].

An alternative method of root canal disinfection was indicated by Tayeb et al. [58]. In animal studies, they demonstrated, comparable to TAP, the antibacterial efficacy of propolis used as an intracanal medicament in REP.

So far, the available reports are dominated by protocols involving two visits with interference in the canal lumen, between which disinfectants are applied for one to four weeks [44]. However, this entails a risk of reinfection of the canal, which is why there are proposals for a single-visit procedure.

One of the alternative methods may be photoactivated disinfection (PAD). It was successfully used in root canal disinfection of mature teeth [59–62]. Attempts are being made to use it in REP [62,63]. It is based on activating a photosensitizer (tolonium chloride or toluidine blue) with a low-power laser light at an appropriate wavelength. The energy released during this process causes the formation of a reactive oxygen species (ROS), which is responsible for the destruction of bacterial cells. This method is highly selective, non-toxic, and does not induce microbial resistance. In addition, the procedure is painless and quick. If necessary, the treatment may be repeated several times during one visit. It also has the advantage of performing the entire REP in a single visit, which minimizes the risk of contamination during subsequent tooth opening [62–64]. However, it requires appropriate instrumentation.

Some researchers reported successful REPs performed in a single visit, using only rinsing disinfectants [65,66].

Based on our initial studies, it should be noted that the phenomenon of extravascular neoangiogenesis was observed in treated and untreated adult teeth (Figure 2). This phenomenon was common when there were processes associated with damage to the tooth wall and pulp as part of the repair processes. Direct angiogenesis from pulp stem cells was not found in the teeth we studied. Furthermore, it is noteworthy that the mechanisms modifying angiogenesis are different in inflammation associated with neutrophil infiltration, compared to lymphocytic infiltration. These two different immunological responses in the first case were associated with vessel formation only after the applied treatment, while in the second case, angiogenesis was present all the time. Similar findings were reported by Meschi et al. [67].

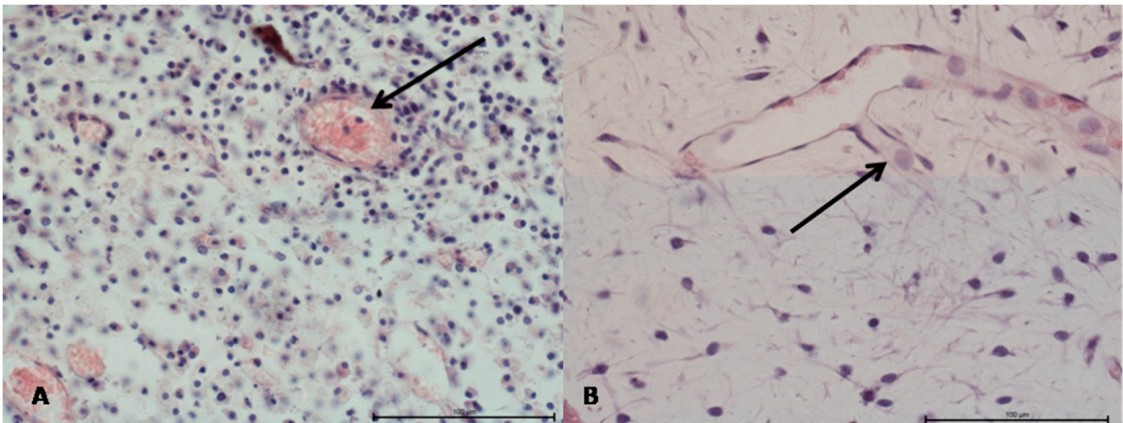

**Figure 2.** Human molar tooth. (**A**) Inflammation of the tooth pulp. Blood-filled blood vessels are visible (arrow), surrounded by a moderate macrophage–lymphocytic infiltrate. (**B**) Angiogenesis (arrow) in human dental pulp (H&E stain enl. 400×).

### 3.2. Components of the Pulp Regeneration Process

A prerequisite for pulp regeneration is the supply of its essential components: stem cells, growth factors, and scaffolds to the canal lumen [8].

Dental stem cells, which are an example of mesenchymal stem cells (MSCs), are found in the dental pulp, dental papilla, periodontium, and periapical region. Many types are identified, which show a high multilinear proliferative potential and the ability to differentiate into cells involved in REP [68–71] (Table 1).

**Table 1.** Stem cells involved in apexogenesis and REP.

| Type | Function |
|------|----------|
| Human Dental Pulp Derived Stem Cells (HDPSCs) | Differentiation toward odontoblasts, osteoblasts, adipocytes, neurons, initiation of angiogenesis |
| Stem Cells from Human Exfoliated Deciduous Teeth (SHEDs) | Differentiation toward odontoblasts, osteoblasts, adipocytes, neurons |
| Periodontal Ligament Stem Cells (PDLSCs) | Osteogenesis |
| Dental Follicle Stem Cells (DFSCs) | Differentiation toward odontoblasts, fibroblasts, osteoblasts, cementoblasts |
| Human Umbilical Vein Endothelial Cells (HUVECs) | Angiogenesis |
| Stem Cells from the Apical Papilla (SCAPs) | Differentiation toward odontoblasts |

Growth factors, isolated from platelets, plasma, dentinal tubules, dental papilla, and the periapical region of the root, are bioactive proteins that act as signaling and regulatory factors. By binding to cell receptors, they influence a number of interactions between the cells and the extracellular matrix (ECM). In addition, they act as regulators of migration, differentiation, cell proliferation, and tissue formation [11,72–77] (Table 2).

**Table 2.** Growth factors involved in apexogenesis and REP.

| Type | Function |
|------|----------|
| Bone Morphogenetic Protein (BMP) | Dentinogenesis |
| Vascular Endothelial Growth Factor (VEGF) | Proliferation, angiogenesis |
| IGF Insulin-like Growth Factor (IGF) | Proliferation |
| TGF-β Transforming Growth Factor-β (TGF-β) | Migration, proliferation |
| FGF Fibroblast Growth Factor (FGF) | Migration, proliferation, dentinogenesis |
| Platelet-Derived Growth Factor (PDGF) | Migration, angiogenesis |

The scaffold creates a suitable environment for the processes mentioned above and provides support for the emerging new tissue [16]. An ideal carrier should show biocompatibility, biodegradability, no cytotoxicity or inflammation induction, adequate porosity, and pore size [9,54]. In regenerative endodontics, the fibers present in blood clot (BC), platelet-rich plasma (PRP), and platelet-rich fibrin (PRF) are employed as a standard scaffold [11]. Reports in which other scaffolds were used are available in the literature, and knowledge on this subject is developing rapidly [71,75,78–85] (Table 3).

The simplest way to obtain a scaffold in the root canal is to induce bleeding from the periapical region, which causes blood to flow into the canal and form a clot. Its main ingredient is fiber, which plays a key role in the formation of new connective tissue. Stem cells from the periapical region of the root also enter the canal along with the blood. However, BC contains few growth factors [18]. Moreover, the procedure requires using an anesthetic without vasoconstrictors, which may be associated with pain during the induction of bleeding. Excessive instrumentation of the periapical region also poses the risk of damaging Hertwig's sheath, resulting in root growth inhibition and procedure failure [86]. In some cases, it can be problematic to obtain sufficient bleeding from the periapical region [87]. In addition, a study by Rizk et al. [88] found that greater crown discoloration and smaller root growth in length and thickness were observed with BC than PRF.

**Table 3.** Selected substances proposed as scaffolds in regenerative endodontics.

| Scaffold | Origin | Chemical Structure | Characteristics |
|---|---|---|---|
| Fibrin | Natural | Protein | Low-cost, biocompatible, derived from blood plasma, does not induce an immune response |
| Silk | Natural | Protein | Biodegradable, biocompatible, does not induce an immune response |
| Chitosan | Natural | Polysaccharide | Biocompatible, biodegradable, may cause allergic reactions |
| Hyaluronic acid | Natural | Polysaccharide | Biocompatible, low immunogenic potential, hydrogel-forming extracellular matrix |
| Collagen | Natural/ synthetic | Protein | Biocompatible, low immunogenic potential, hydrogel-forming extracellular matrix |
| Self-assembling peptides | Synthetic | Peptides | Biocompatible, forming hydrogels |
| Polylactic acid (PLA), Polyglycolic acid (PGA), Poly(lactide-co-glycolide) (PLGA) | Synthetic | Polyesters | Biocompatible, biodegradable, may cause slight inflammatory reactions |
| Bioactive ceramics | Synthetic | Calcium phosphates, Bioactive glasses (mixture of sodium silicon oxides, calcium, magnesium, iron, etc.) | Biocompatible, low immunogenic potential, osteoinductivity |

PRP, the platelet concentrate obtained from patient plasma, acts as both a scaffold and a rich reservoir of cytokines and growth factors released from platelet *α* granules [89]. PRF is a newer substance, a so-called second-generation platelet concentrate, containing a fibrin membrane enriched in platelets, growth factors, and cytokines [90].

Limitations of the PRF or PRP method are the need for venous blood sampling, which can be problematic in young patients, the need for specialized equipment, and difficulty in applying the substance to the canal due to its gelatinous consistency [88,91]. Therefore, the evaluation of the efficacy of BC, PRP, and PRF in REP is challenging.

Shivashankar et al. [18] found comparably high efficacy of PRP, PRF, and BC in stimulating root growth, while PRP was best for healing the periapical region. Narang et al. [92] came to a different conclusion, observing better efficacy of PRF than PRP and BC in promoting healing of periapical lesions. The high efficacy of PRF in REP and treatment of inflammation was also confirmed in other reports [72,91,93,94].

### 3.3. Root Canal Lumen Closure

Proper sealing of the canal lumen, which prevents reinfection, is one of the key factors necessary for successful REP. For this purpose, MTA, Biodentine, and types of glass ionomer cement (GIC) are used. MTA is a biocompatible material that exhibits odontotropic properties, chemically bonds to dentin [95], and can bond in moist environments [18]. Peng et al. [96] demonstrated a higher risk of developing periapical tissue infection when using GIC with a REP procedure, compared to MTA. In order to minimize the risk of crown discoloration after applying MTA, the use of its white variant or Biodentine is recommended [97,98]. Additionally, Biodentine has the advantage of exhibiting similar mechanical properties to dentin and very low cytotoxicity [44]. Many authors point out the synergistic effect of MTA or Biodentine and PRF in promoting the treatment of periapical lesions of immature teeth [26,94,98,99] and stimulating stem-cell differentiation [100]. In vitro studies by Woo et al. [77] confirmed the efficacy of a combination of PRF and MTA preparations in stimulating stem cells toward odontoblastic differentiation. Before using the sealant, it is advisable to apply a collagen membrane to facilitate accurate applica-

tion, especially in wide canals, and prevent sealant displacement [54]. The appointment concludes with the placement of a hermetic permanent filling.

### 3.4. Follow-Ups

The patient should be followed up every three months during the first year, every six months for two years, and then every 12 months for five years. Any inflammatory features should be assessed at each visit through an interview, physical, and radiological examinations [44].

The main signs of successful REP are the absence of inflammation, healing of inflammatory lesions in the periapical tissues, increased root length and wall thickness, lack of external inflammatory resorption, and a positive response to the pulp vitality test [45]. In practice, the results of REP treatment are variable. Chen et al. [81] described possible five types of response to REP: Type 1—root wall growth in length and thickness and closure of the apical opening; Type 2—no visible root growth in thickness and length, but closure of the apical opening can be observed; Type 3—root growth in length and thickness, no closure of the apical opening; Type 4—obliteration/calcification of the root canal lumen; Type 5—a barrier formed by the hard tissue between the root apex and the applied clot sealing material (MTA/Biodentine).

The most substantial evidence of living tissue in the canal is the progression of root development visible on the X-ray, which is not always confirmed by pulp vitality tests [1]. The reason for this phenomenon may be the thickness of the MTA layer and its position below the enamel–dentin junction, influencing the negative response, despite the presence of living tissue [18].

### 3.5. Comparison of Methods and Effects of REP

Many REP case reports can be found in the literature, differing in the details of the procedure. The examples of methods and effects of REP treatment of immature teeth with non-vital pulp reported by different authors are presented below and in Table 4.

**Table 4.** Selected REP methods used by different authors.

| Year | Author | Irrigation/Disinfection | Scaffold | Restoration |
|---|---|---|---|---|
| 2007 | Thibodeau and Trope | 1.25% NaOCl, TAP | BC, MTA | Composite |
| 2012 | Aggarwal et al. | 5.25% NaOCl, distilled water, 2% CHX, TAP, Ca(OH)$_2$ | BC, MTA | GI, composite |
| 2013 | Jadhav et al. | 2.5% NaOCl, TAP | BC, PRP, collagen | RMGI |
| 2014 | Johns et al. | 5.25% NaOCl, 0.9% NaCl, PAD | PRF, MTA | Composite |
| 2015 | McCabe | 5% NaOCl, 17% EDTA | BC, MTA | GI |
| 2016 | Topçuoglu et al. | 2.5% NaOCl, 0.9% NaCl, 17% EDTA | PRP, Biodentine | Composite |
| 2017 | Shivashankar et al. | 5.25% NaOCl, TAP | BC/PRP/PRF, MTA | IRM |
| 2018 | Adhikari and Gupta | 3% NaOCl, 17% EDTA, Ca(OH)$_2$ | PRF, MTA | Composite |
| 2019 | Rahim et al. | 1.5% NaOCl, 0.9% NaCl, PAD, 17% EDTA | BC, MTA | GI, composite |
| 2020 | Maniglia-Ferreira et al. | 2.5% NaOCl, 17% EDTA, 2% CHX, DAP, Ca(OH)$_2$, zinc oxide | BC, MTA | RMGI |
| 2020 | Elfrink et al. | 2% NaOCl, DAP/TAP | BC, MTA | Composite |
| 2020 | Sakthivel et al. | 0.5% NaOCl, 17% EDTA, Ca(OH)$_2$ | PRF, BC, Collagen, MTA | GI, composite |

One of the first described cases of successful REP took place in 2007 [53]. At that time, 1.25% NaOCl and TAP were used to disinfect the canal, and BC was used as a scaffold, subsequently covered with MTA. Follow-up appointments were arranged after 3, 6, 9 and 12 months—the tooth remained asymptomatic, including normal response to percussion

and palpation tests, normal mobility but no response to electrical pulp vitality tests. The radiological image revealed the continuation of root development, closure of the root apex, and growth of the root wall in thickness.

In 2012, Aggarwal et al. [101] reported a case comparing a standard $Ca(OH)_2$ apexification procedure with REP in the same patient in two different teeth. For REP, 5.25% NaOCl and 2% CHX, TAP, and $Ca(OH)_2$ were used during disinfection. In addition, BC and MTA were applied as scaffolds. Follow-up visits took place after 3, 6, 12, 18, 24 months. Both teeth were asymptomatic and showed no pathological mobility; an additional increase in root wall length and thickness and closure of the apical foramen were observed in the tooth after REP.

In 2013, Jadhav et al. [102] described three REP cases, using BC or PRP as cellular scaffolds. The disinfection protocol used 5.25% NaOCl and TAP in each case, and the teeth were closed with a resin-modified glass ionomer (RMGI). Follow-up appointments were arranged after 6 and 12 months. The teeth remained asymptomatic; radiographs showed root growth in length and thickness and closure of the apical opening. Despite similar treatment effects with a blood clot alone and platelet-rich plasma, in addition, the authors recommend the use of PRP and collagen membrane for additional clot stabilization, particularly in cases where inducing the apical region bleeding is not possible.

In 2014, Johns et al. [62] performed a single-visit REP in two upper central incisors in a nine-year-old boy. First, the canals were rinsed with 5.25% NaOCl and then PAD was used. After disinfection, PRF and MTA were applied. During follow-up visits after 6 and 10 months, further root development and retreatment of periapical inflammatory lesions were observed. Root formation was finally achieved after 10 months, but the response to vitality tests remained negative.

In 2015, a case of successful REP using a single-visit protocol was reported by McCabe [66]. Ultrasound-activated 5% NaOCl was used for disinfection, and 17% EDTA was the final irrigant. After bleeding of the apical region, MTA was applied, and the cavity was restored with GI. Follow-up visits were made after 6 weeks and 3, 6, 12, and 18 months. The tooth remained asymptomatic. Radiographs showed root wall growth, which manifested itself in an increase in its length and thickness and closure of the apical opening.

In 2016, Topçuoglu et al. [65] attempted to treat three immature mandibular molars with pulp necrosis in children aged eight and nine. The procedure was performed in a single visit. After disinfection with 2.5% NaOCl, PRP was applied to the canals, which were then closed with Biodentine, and the cavities were then filled with composite. Further root development and an absence of pathological symptoms were observed.

In 2017, Shivashankar et al. [18] conducted a comparative analysis of the efficacy of using PRP, PRF, and BC. Twenty immature teeth with necrotic pulp were included in each group. The canals, after careful instrumentation, were rinsed with 5.25% NaOCl and filled with TAP for three weeks. At the next visit, PRP, PRF, or BC was applied to the canal. The teeth were closed with intermediate restorative material (IRM). After 3, 6, 9, and 12 months of observation, none of the patients showed any pathological clinical symptoms or the development of new periapical lesions or an increase in its size. The highest healing efficiency of periapical lesions was recorded after PRP application.

In 2018, Adhikari et al. [103] published a REP case in which 3% NaOCl and 17% EDTA were used in the irrigation protocol and $Ca(OH)_2$ was an intracanal medication. PRF and MTA were utilized as a cellular scaffold, and the tooth was closed with composite resin. Follow-up visits at 1, 6, and 12 months showed healing of the periapical lesions, growth of the root wall in thickness, and closure of the apical opening. After 12 months, the tooth had to be extracted due to fracture; it underwent histopathological examination, which revealed cementum-like tissue on the root canal walls.

In 2019, Rahim et al. [63] performed a single-visit REP in the central upper incisor. Canal disinfection was performed using 1.5% NaOCl and the PAD method. Seventeen percent EDTA was used as the last irrigating solution. After bleeding of the periapical region, the clot was covered with a collagen membrane and MTA. Follow-up visits were

performed after 3, 6, 9, and 12 months. Complete root development was observed after 12 months of follow-up.

In 2020, Maniglia-Ferreira et al. [104] described a case of REP treatment of two central incisors with a 12-year follow-up. It consisted of 2.5% NaOCl, 17% EDTA, and 2% CHX gel used for disinfection (for one week), and then DAP was applied to one tooth and Ca(OH)$_2$ to the other (for three weeks). BC and MTA were employed as a scaffold, and the teeth were restored with RMGI. The patient reported no pain for 12 years, no reaction to palpation or percussion, and cone beam computed tomography (CBCT) revealed root growth in length and thickness, increased calcification, and a root shape defect. The authors emphasized that there was no mention in the available literature of possible tissue invagination during root formation due to REP. As a probable cause, they suggested the compression or stretching of Hertwig's sheath as a result of trauma, which led to pulp necrosis in the examined tooth.

In 2020, Elfrink et al. [105] published a study of 47 teeth undergoing REP, which was carried out between 2009 and 2017. Of the 47 evaluated cases, only in three, the procedure failed. For disinfection, 2% ultrasound-activated NaOCl and, initially, TAP was used, with DAP after 2015. BC with MTA was used as a carrier. Teeth were restored with composite resin. Cases of ankylosis were documented in teeth undergoing the procedure, but this could be related to the trauma that led to the endodontic treatment. Discoloration of tooth crowns was noted, probably due to obliteration, the use of minocycline, or MTA. To minimize the risk of discoloration, the use of DAP and Biodentine was suggested.

In 2020, Sakthivel et al. [72] described a case of treatment of post-traumatic pulp necrosis, complicated by periapical tissue inflammation in a medial upper incisor. Rinsing agents included 0.5% NaOCl and 17% EDTA. Calcium hydroxide was placed as an intracanal dressing paste for two weeks. At the next visit, the periapical area was bled to obtain stem cells, and PRF was utilized as a scaffold and source of growth factors. The canal was closed with MTA placed on a collagen membrane. Follow-up visits were made after two weeks, then after 1, 3, 6, 12, and 24 months. After two years of follow-up, healing of the periapical lesion and closure of the apical foramen were recorded.

### 3.6. Proposed Protocol of REP

As it can be inferred from this review, many REP protocols have been proposed, and no single optimized algorithm has emerged. We developed a procedure that is based on the literature and constitutes a compromise between safety and feasibility. Therefore, the solutions reported in the literature [18,62,66,101] that are debatable, such as irrigation with high concentrations of NaOCl, were not included. TAP showed high efficacy against pathogens present in the root canal [40,41] and is, therefore, recommended in many protocols [18,35,101,105]. However, it shows potential side effects, such as the risk of developing bacterial resistance, coronal discoloration, cytotoxicity, and difficulty in removal from the root canal [40,41]. Thus, it seems that using calcium hydroxide as a dressing with proven antimicrobial efficacy, which is also readily available and widely used in dentistry, is a better option [44]. So far, despite the ongoing debate, the superiority of any of the REP methods used (BC, PRP, PRF) is yet to be unequivocally demonstrated [18]. Blood clot formation is a simple method that does not require drawing blood from the patient or having specialized equipment, and thus can be performed in any dental office. The details of the proposed protocol are presented below (Table 5).

The first visit: After local anesthesia, isolation with a rubber dam and disinfection of the operating field with iodine solution, endodontic access should be performed and the tooth chamber secured with a bonding system or Biodentine material (if TAP is used). Irrigation of the canal can be achieved with a 2% NaOCl solution (20 mL, 5 min), followed by 17% EDTA (20 mL, 5 min) and the same amount of 0.9% NaCl. The endodontic needle should be placed 1–2 mm from the apex and the working length determined based on the intraoperative radiograph and confirmed by endometer indication. The effect of NaOCl can be enhanced by applying ultrasound or by heating the solution to 37 °C. After drying the canal with sterile paper points, calcium hydroxide should be applied intracanally. The

canal should be secured with a sterile cotton pad/sponge/Teflon, and the tooth should be sealed, e.g., with glass ionomer cement. It is recommended to monitor the entire procedure using magnification, working with a dental microscope.

**Table 5.** REP—proposed protocol.

| First Appointment | Second Appointment (Only in the Absence of Signs of Inflammation) | Follow-Up (Clinical and Radiological Examination) |
|---|---|---|
| 1. Local anesthesia (may be done with vasoconstrictor) | 1. Local anesthesia (3% mepivacaine without vasoconstrictors) | 1. After three months |
| 2. Isolating the operating field with a rubber dam, disinfecting with iodine solution (Povidone/Betadine) | 2. Isolating the operating field with a rubber dam, disinfecting with iodine solution (Povidone/Betadine) | 2. After six months |
| 3. Endodontic access (sterile drills) | 3. Reopening of the tooth (sterile drills) | 3. After nine months |
| 4. Intra-chamber application of a bonding system or Biodentine * <br> * If TAP is used | 4. Irrigation with 20 mL of 0.9% NaCl (5 min), 20 mL of 17% EDTA (5 min) | 4. After 12 months |
| 5. Irrigation with 20 mL of 2% NaOCl (5 min, ultrasound activation), 20 mL of 17% EDTA (5 min), 20 mL of 0.9% NaCl (5 min) | 5. Draining the canal with sterile paper points | 5. Every six months |
| 6. Draining the canal with sterile paper points | 6. Inducing bleeding of the apical area * (e.g., sterile, #25 K-file, 2 mm beyond the apex), clot formation (15 min), collagen membrane application <br> * Or PRP/PRF application | |
| 7. Calcium hydroxide application (or TAP/DAP—not recommended) | 7. Application of 3–4 mm MTA/Biodentine to the clot | |
| 8. Tight temporary filling, e.g., RMGI | 8. Tight permanent filling (RMGI, composite) | |

The second visit should take place after two to four weeks. If symptoms suggesting an ongoing inflammatory process persist or occur, the protocol from the first visit should be repeated. In the absence of symptoms, one can proceed to the final REP. For this purpose, the patient should be anesthetized (vasoconstrictors are not recommended to facilitate bleeding), the surgical field isolated and disinfected, and the temporary filling removed. Irrigation should be carried out with 0.9% NaCl (20 mL, 5 min) to wash out any residual TAP from the canal completely. Rinsing should be completed with 17% EDTA (20 mL, 5 min). After drying the canal with sterile paper points, bleeding of the apical region must be induced, e.g., by inserting a sterile, bent #25 K-file 2 mm beyond the apex and then waiting for clot formation for approx. 15 min. The clot should be located approx. 3 mm below the CEJ. An alternative method is to use PRP or PRF. Additionally, a collagen membrane can be used to help to stabilize the clot. The clot should then be covered with approximately 3–4 mm of MTA/Biodentine and sealed, e.g., using RMGI and composite. If MTA is used, the final filling is possible on the next visit. Finally, it is necessary to close the canal tightly and restore the tooth crown.

Follow-ups: Follow-up appointments, including clinical and radiological examination, should take place every 3 months during the first year, then every 6 months for two years, and then every 12 months for five years.

## 4. Discussion

REP probably represents the future of treatment of immature permanent teeth with pulp necrosis. The undoubted advantage over the standard apexification procedure using calcium hydroxide and MTA is the possibility of further root development, its growth in length and thickness, and closing of the apical foramen. The fundamental prerequisite for the success of the procedure is the correct qualification of the patient, effective disinfection

of the root canal, and tight closure of the cavity. Despite intensive research carried out worldwide, this procedure still requires refinement and standardization of the treatment protocol. There are also some limitations, such as the lengthy treatment time and the need for the patient to attend numerous follow-up appointments. Complications, such as crown discoloration, tooth fracture, ankylosis, calcification/obliteration of the root canal lumen [89], periapical tissue inflammation, or lack of further root growth, can also be expected in some cases [106]. The use of calcium silicate-based cements (CSCs), such as MTA or Biodentine, was shown to contribute to crown discoloration, with Biodentine contributing significantly less than MTA. The mechanism of action for MTA is associated with the oxidoreductive potential of bismuth oxide, which, when in contact with strong oxidizing agents, such as sodium hypochlorite or collagen, causes tooth discoloration. In addition, direct contact of MTA with blood was shown to exacerbate discoloration. Another disadvantage of MTA is its long setting time (2 h and 45 min), which increases its porosity and, as a consequence, increases blood absorption and causes greater discoloration of the tooth crown. The prevalence of Biodentine over MTA is undoubtedly due to the lack of bismuth oxide in its composition, a much shorter setting time (12 min), and a lower rate of tooth crown discoloration. There are also reports in the recent literature of other CSCs, also with shorter setting times than MTA, which may also be good alternatives; these include, for example, TotalFill or PCM. Other types of MTA that do not contain bismuth oxide are also available, such as Neo MTA or Fillapex MTA [107–109]. Tight restoration of the tooth after treatment is also fundamental to the success of REP. An effective bond between the biomaterial used to close the root canal and the final tooth filling is essential. The goal is to perform the final restoration of the tooth in a single visit, which not only increases the chance of therapeutic success, but also helps to reduce the costs of the procedure. Palma et al. [110] compared the effects of Biodentine, TotalFill, and PCM on shear bond strength for immediate and delayed composite restorations. This study demonstrated the superiority of Biodentine and TotalFill over PCM. According to some reports, there is no unequivocal evidence of a significantly higher chance of long-term success in treating immature teeth with REP, compared to classic apexification [19,111,112]. At the moment, it is also difficult to univocally assess the efficacy of this method, as most of the reports available in the literature describe successful cases, whereas the therapeutic failures are omitted [25]. From a practical point of view, publications discussing possible failures, with analysis of their causes and proposals for their prevention, would be instrumental. So far, there are few systematic reviews and papers based on a large number of cases, which provide a more comprehensive picture of the discussed issue. The solution could be large, multicenter clinical trials with long-term follow-ups to evaluate the long-term efficacy of this method. Nevertheless, applying tissue engineering methods seems promising. The dissemination of this method and the advancement of knowledge may also contribute to the use of REP in immature molar teeth. In the long term, it may be possible to extend the use of regenerative endodontics also to mature teeth [113,114].

## 5. Conclusions

This literature review suggests that REP may become the standard for treating immature permanent teeth with necrotic pulp in the near future. The formation of living tissue in the root canal with stem cells, growth factors, and scaffolds is a milestone in endodontics. However, this method requires further intensive research, due to many controversial issues regarding clinical management and relatively poorly documented efficacy. In addition, it is necessary to develop an optimal management protocol in specific cases. The need for comparative analysis of the long-term effectiveness of REP and apexification is also indicated.

**Author Contributions:** Conceptualization, J.Z., K.H., and P.K.; validation, M.D.; formal analysis, K.H., M.D.; investigation, J.Z., K.H., and P.K.; resources, M.D.; data curation, J.Z., K.H., and P.K.; writing—original draft preparation, J.Z.; writing—review and editing, K.H., P.K., and M.D.; visualization, P.K.; supervision, M.D.; project administration, M.D. All authors have read and agreed to the published version of the manuscript.

**Funding:** This work was financed by a grant from Wroclaw Medical University, number SUB.B180.21.055.

**Institutional Review Board Statement:** Not applicable.

**Informed Consent Statement:** Not applicable.

**Conflicts of Interest:** The authors declare no conflict of interest.

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
