# Peer review of "Regenerative Endodontics as the Future Treatment of Immature Permanent Teeth"

_applsci, doi:10.3390/app11136211_

Round 1

Reviewer 1 Report

This research is under the scope of this journal; the topic is relevant for readers, and this research deals with potentially significant knowledge of the field.

However, there are some concerns about the present manuscript: 

Abstract

  • Keep the same terminology, as Regenerative endodontic Procedure (REPS) ... and standardize throughout the text with the REPs, avoid alternating with revascularization ... as it can create confusion in readers.

Introduction

  • Page 2 line 58 – F. Bachns and  M. Trope corect for  Banchs. Please, correct typos in all manuscript.
  • The Dental papilla, when the root is formed, is in the apical zone, called the papilla apical (stem cells from apical papilla - SCAPs) that remain until the apical closure of the root. Also these, cells can the survival of SCAPs at the infection of the tooth, (in REPs animal study and a Clinical Case, read these references (doi.org/10.3390/APP9193942 and DOI: 10.1016/j.joen.2017.03.005). HERS and the apical papilla are two embryologic structures that coordinate all the radicular development through epithelial-mesenchymal interactions. Apical papilla was a reservoir for mesenchymal stem cells (SCAPs) fundamental for the root development of immature teeth. Normal dentin–pulp complex development requires not only the survival of HERS and ERM but also SCAPs. Endodontic therapies applied to immature teeth that do not affect the apical papilla viability are a key determining factor, guiding successful root development.
  • The aim: This  Study (doi.org/10.3390/APP9193942) also compare the “protocols” suggests for the AAE and ESE. What is the importance of this review for the clinical? You do not think this study is included in the others already done? What is the news in this narrative review?

M&M

  • This section would better communicate to readers if restructured. A flowchart or diagram of the selected article processing would be valuable. Strategy search: you must present a full electronic search strategy for at least one database, including any limits used, such that it could be repeated. Made a flowchart, to explain to reads the sequence of the narrative study, and with the inclusion, exclusion criteria for the selection of these articles.

Discussion 

- it seems too short to discuss so relevant matter of REPs!! 

- Please discuss the Impact of hydraulic calcium silicate-base cement on discolouration and the lower values for a shear bond. Please read these articles, since this type of calcium silicate cement is associated with colour change in the medium / long term (https://doi.org/10.3390/app10175793, 10.1016/j.joen.2017.04.002). The ProRootMTA (Portland base of hydraulic Calcium silicate-based cement) had a contrast agent (oxide bismuth) with a big effect on discolouration https://doi.org/10.1371/journal.pone.0240634. The strategy on the part of the reviewer is to increase the quality of the presentation of these topics. Overall, the rest of the content overcomes this limitation. Other drawbacks of MTA were a lowers values shear bond. One of the major problems with MTA hydraulic cement, in addition to its setting time, is the weak connection to restorative materials, with very low values and unpredictable connections to restoration material (Palma, Materials MDPI, 2018). Also, I suggest reading “Effect of restorative timing on shear bond strength of composite resin/calcium silicate-based cement adhesive interfaces. Clin Oral Invest (2020). https://doi.org/10.1007/s00784-020-03640-7)”. 

-What is the new in this protocol propose? Discuss with the other proposal/literature.

  • Please, clarified other limitations of this study?
  • And, clarified the future perspectives.

Conclusions

  • The conclusion section should summarize the results of this narrative review. 

References

  • But references are not standardized. The references need to be inserted correctly, some time is before the endpoint and others are after. 
  • And when you had in the text the “authors et al.” references should come immediately afterwards, not in the final of the sentence.
  • The titles of references have a different format, 
    the title of the article is written in capital letters at the beginning of words, others only in lower case. Also, the standardized format of presentation in the journal's name. Because names have written in a different format, one is not abbreviated, others are not.

Reviewer 2 Report

Dear Authors, this is an interesting paper, but it requires some major revisions before it is published.

1) English language should be improved - at some points articles are missing, also some words sound not familiar. Please, check it carefully once again

2) If this is a review, in the "Materials and methods" section, you must provide the readers with the information on how (methods) you chose the materials (ie the manuscripts) - there is no way I can tell why you chose 1 article from each year - does it mean only 1 procedure was performed each year? Or was only 1 successful? (of course not - but I just want to draw your attention to what the readers' impressions may be)

3) lines 48-50 - calcium hydroxide does not increase the risk of fracture per se (https://www.jendodon.com/article/S0099-2399(17)31117-2/abstract)

4) lines 45-50 - it is written in a little bit unclear manner - MTA/Biodentine co not need to be replaced, and they constitute the barrier that is developed over time when calcium hydroxide is used. Please rewrite it in a way that is going to be more structured and repeatable.

4) lines 75-77 - rewrite it

5) lines 117-118 - please elaborate on cytotoxicity of NaOCl

6) lines 123-124 - this L-alpha-lecitin is not used (and I guess is not going to be used) in clinical practice. This is irrelevant.

7) lines 156-166 - you elaborate on the properties of TAP but there is little information regarding calcium hydroxide. Ca(OH)2 has some positive properties - eg it increases the release of growth factors and proteins from radicular dentine. Please search for that information and include that in the text.

6) lines 187-196 - this fragment needs to be rewritten - what studies? Tell us a bit more about them. 

7) Table 3 - please rearrange it so that it is more comprehensible.

8) lines 405-419 - please follow the guidelines of ESE. TAP is no longer advocated as the dressing of choice, the period between the 1st and the 2nd appointment should be 2-4 weeks, not 1-4 weeks. You are not in position (you're not a scientific association) to provide strict guidelines - as researchers you can provide some instructions on how to do it but do not undermine the position statements of eg ESE.

In conclusion, the text needs to be rewritten in a structured, more comprehensible way. Please take a closer look at other reviews and try to conform to those rules a bit more and it is going to be fine.

Round 2

Reviewer 1 Report

This research is under the scope of this journal; the topic is interesting for readers and this research deals with potentially significant knowledge to the field and an open new way for future studies.

The authors improved the quality of the manuscript after the reviewer's indications.

Author Response

We would like to thank you so much for your time and for your valuable feedback which helped us to improve this paper.

Reviewer 2 Report

Congratulations, it is much better now. Please describe the flowchart in the text. Also, there are some minor stylistic issues:

Line 1 standard->conventional

Line 61-62 – is currently attracting-> currently attracts

Line 66: scaffolding -> a scaffold

Line 69 living -> vital

Line 86: with the apical opening -> if the diameter of the apical foramen is…

Line 133: boosts the bactericidal effect and negatively affects cell proliferation -> improves the bactericidal effect, but at the same time hinders cell proliferation

Line 142: EDTA use -> the use of EDTA

„there are also reports of 153 potential disadvantages of TAP, such as” -> “Several disadvantages were also reported when TAP was used, such as…”

Line 180: tooth chamber->pulp chamber

Line 230: belong->are an example of

Line 241: absence of-> no cytotoxity

Line 244: There are reports of attempts to use many -> Also, reports, in which other scaffolds were used, are available in literature,

 Line 247: induce bleeding from the periapical…

Line 352: showed root wall growth in length and… -> showed root wall growth which manifetsted itself in an increase in its length and…

Line 365: appearance or enlargement -> development of a new periapical lesion or its increase in size

Line 377: were made -> were scheduled/performed

Line 411: “Based on the literature, we developed a procedure” ->  We developed a procedure that is based on the literature and constitutes a compromise between safety and feasibility”

Line 420: the superior efficacy -> superiority

Line 470: should -> can in some cases

Line 482: TotalFill, PCM, or FKG Dentaire – please check it: FKG dentaire is the name of the company, whereas totalfill is the name of the product

Author Response

Dear Reviewer,

We would like to sincerely thank you for your time, for your very thorough and insightful review and for your valuable suggestions which helped us to improve this article.

The flowchart has been described in the text as well as minor stylistic issues have been corrected.